# Mucopolysaccharidosis IVA: Current Disease Models and Drawbacks

**DOI:** 10.3390/ijms242216148

**Published:** 2023-11-09

**Authors:** Andrés Felipe Leal, Carlos Javier Alméciga-Díaz, Shunji Tomatsu

**Affiliations:** 1Nemours Children’s Health, Wilmington, DE 19803, USA; andres.lealbohorquez@nemours.org; 2Institute for the Study of Inborn Errors of Metabolism, Faculty of Science, Pontificia Universidad Javeriana, Bogotá 110231, Colombia; cjalmeciga@javeriana.edu.co; 3Faculty of Arts and Sciences, University of Delaware, Newark, DE 19716, USA; 4Department of Pediatrics, Graduate School of Medicine, Gifu University, Gifu 501-1193, Japan; 5Department of Pediatrics, Thomas Jefferson University, Philadelphia, PA 19144, USA

**Keywords:** chondrocytes, fibroblasts, mouse, mucopolysaccharidosis IVA, rat

## Abstract

Mucopolysaccharidosis IVA (MPS IVA) is a rare disorder caused by mutations in the N-acetylgalactosamine-6-sulfate-sulfatase (*GALNS*) encoding gene. GALNS leads to the lysosomal degradation of the glycosaminoglyccreasans keratan sulfate and chondroitin 6-sulfate. Impaired GALNS enzymes result in skeletal and non-skeletal complications in patients. For years, the MPS IVA pathogenesis and the assessment of promising drugs have been evaluated using in vitro (primarily fibroblasts) and in vivo (mainly mouse) models. Even though value information has been raised from those studies, these models have several limitations. For instance, chondrocytes have been well recognized as primary cells affected in MPS IVA and responsible for displaying bone development impairment in MPS IVA patients; nonetheless, only a few investigations have used those cells to evaluate basic and applied concepts. Likewise, current animal models are extensively represented by mice lacking GALNS expression; however, it is well known that MPS IVA mice do not recapitulate the skeletal dysplasia observed in humans, making some comparisons difficult. This manuscript reviews the current in vitro and in vivo MPS IVA models and their drawbacks.

## 1. Introduction

Mucopolysaccharidosis IVA (MPS IVA) or Morquio A syndrome is a lysosomal storage disorder (LSD) caused by mutations in the N-acetylgalactosamine-6-sulfate-sulfatase (*GALNS*) encoding gene [1,2,3]. Under physiological conditions, the GALNS enzyme leads to lysosomal degradation of the glycosaminoglycans (GAGs), keratan sulfate (KS), and chondroitin 6-sulfate (C6S). Impaired GALNS activity results in patients suffering from skeletal abnormalities, short stature, hearing loss, corneal clouding, and pulmonary and cardiac dysfunctions [4,5,6,7,8].

According to the Human Gene Mutation Database (HGMD^®^)’s website, 354 mutations have been reported for the *GALNS* gene (Figure 1). Even though some mutations have been strongly related to MPS IVA severity, a clear genotype-phenotype correlation does not exist [9,10]. Mostly, those mutations expose hydrophobic GALNS core (>70%), thus affecting GALNS folding [11]. Mutations can also affect residues at the active site or even introduce premature stop codons [11,12,13,14,15,16].

Currently, only the use of the elosulfase alfa (Vimizin^®^), a recombinant human GALNS enzyme (rhGALNS), as an enzyme replacement therapy (ERT) is approved for treating MPS IVA patients [18]. ERT is dosed at 2 mg/kg/week in all MPS IVA patients [19,20]. Even though the ERT leads to a significant improvement in the 6 min walk test (6MWT) and slight recovery of the left ventricular ejection fraction [19,21,22,23,24,25], it has been noticed that avascular zones as the growth plate are poorly impacted by ERT [21], suggesting a limited effect on major clinical findings of MPS IVA patients [26]. ERT does not decrease some well-known pathophysiological events, such as oxidative stress [27]. Besides, several alternatives, including pharmacological chaperones [28], substrate degradation enzyme therapy [29], hematopoietic stem cell transplantation [5], and gene therapy [30,31,32,33] have also been evaluated as potential therapeutical options. These strategies are summarized in Figure 2.

All the strategies mentioned above have been evaluated on in vitro models of the MPS IVA, such as skin fibroblasts [30,32,33,34,35] and chondrocytes [36,37,38] or in vivo models as mice [29,31,39] and most recently rats [40]. An overview of these models is displayed in Table 1 and will be explored in detail in upcoming sections.

These MPS IVA models have led to the discovery and testing of promising drugs; nevertheless, some limitations persist. This review explores these models and discusses their advantages and challenges for modeling disease and relevance for exploring new therapeutical approaches.

## 2. In Vitro MPS IVA Models

Even though in vitro approaches are supposed to be the easiest way to explore molecular pathogenic mechanisms and new drug discovery, several key factors should be considered, including culture conditions, supplements, and passages as critical factors affecting cell homeostasis and in vitro behavior. For MPS IVA, skin fibroblasts represent the most common in vitro model; however, chondrocytes and leukocytes have also been reported (Table 2).

### 2.1. Skin Fibroblast as Models of MPS IVA

Skin fibroblasts are well documented in several studies regarding the MPS IVA mechanisms [44]. Classical lysosomal accumulation, lysosomal-related pathway impairment (i.e., autophagy), and pro-oxidant profile are well-documented in fibroblasts [28,30,32,33,44]. These cellular findings are pivotal for screening potential drugs and their ability to rescue the phenotype to wild-type levels. For example, some small molecules with pharmacological chaperone (PC) activity, such as ezetimibe (β-lactam indicated for hypercholesterolemia) and pranlukast (cysteinyl leukotriene receptor-1 antagonist used for chronic bronchial asthma) were reported as potential PCs for GALNS enzymes in MPS IVA fibroblasts. Those PCs led to the rescue of GALNS activity and the recovery of the lysosomal mass and the autophagy efflux [28]. Similar outcomes were reported for these biomarkers upon lentiviral and adeno-associated viral (AAV) gene therapy [33,38,43].

Even though fibroblasts are primarily used for evaluating novel therapeutics in vitro, they have also been used for uncovering basic processes underlying the pathophysiology of the MPS IVA beyond the mutations in the *GALNS* gene. For instance, proteomic analysis using human MPS IVA fibroblast has revealed profound changes in the proteomic profile of organelles such as the mitochondria. In this sense, Alvarez et al., 2019, showed a significant decrease in proteins involved in redox homeostasis and mitochondria-lysosome interplay [35], which confer pathophysiological evidence of the MPS IVA pathogenesis and explain some features such as the predominant pro-oxidant profile described in MPS IVA patients [27,45]. This oxidative profile has also been recognized in skin MPS IVA fibroblast [30,32]. In this sense, we have reported a significant mitochondrial-dependent oxidative stress in human MPS IVA regardless of the mutation in the *GALNS* gene [32], which supports the findings described by Alvarez et al., for the mitochondrial-lysosome crosstalk. Interestingly, by using the CRISPR/Cas9 system to knock in a wild-type GALNS cDNA into the AAVS1 *locus* in human MPS IVA fibroblasts, we found significant recovery in the pro-oxidant profile of the human MPS IVA fibroblasts, one month post-treatment [30,32], suggesting a promising strategy able to overcome the poor impact of the ERT on the oxidative stress in patients [27].

### 2.2. Peripheral Leukocytes

Proteomic profiles from peripheral leukocytes isolated from MPS IVA patients and healthy individuals were recently reported [34]. In those approaches, authors reported 91 and 73 differentially down and upregulated proteins in MPS IVA leukocytes compared to healthy controls. Interestingly, authors found downregulation of some critical proteins linked to glucose metabolism (i.e., glucose-6-phosphate isomerase (G6PI), phosphoglucomutase (PGM1), and glucose-6-phosphate dehydrogenase (G6PD)) which were normalized in peripheral leukocytes from MPS IVA patients under ERT compared to untreated patients [34]. Similar findings were also reported for proteins involved in the lysosomal membrane repair, galectin 3 (LEG3), and the vacuolar protein sorting 35 (VPS35). On the other hand, authors found up-regulation of the vitronectin (VTNC), a glycoprotein that binds GAGs, which could be a physiological response to the increased KS and C6S accumulation. The proteomic analysis also showed the upregulation of the oxidative stress-related proteins neutrophil defensin 3 (DEF3) and lactotransferrin (TRFL) in leukocytes from untreated patients, which surprisingly were normalized in leukocytes from MPS IVA patients under ERT [34], even though some studies suggest that ERT does not improve the pro-oxidant profile in MPS IVA patients [27]. This apparent discrepancy supports the need to explore, in detail, the cellular consequences of the KS and C6S accumulation and the impact on the cell homeostasis of current and upcoming therapies. The authors also found several downregulated enzymes linked to Kreb’s cycle, providing further evidence of the mitochondria–lysosome pathway disturbance in MPS IVA leukocytes as described for MPS IVA fibroblasts [35]. These findings should drive upcoming research to understand MPS IVA not only as a lysosomal-affecting disease but as a cell-homeostasis-affecting pathology. Finally, dysregulation was also observed for the myeloblastin (PRTN3, up-regulated) and coronin-1A (CORO1A, down-regulated), which are related to the type I and II collagen degradation, providing new molecular insights associated with the well-described collagen homeostasis disturbance in MPS IVA patients [46,47].

### 2.3. Chondrocytes

Although MPS IVA patients primarily suffer from skeletal dysplasia, most in vitro modeling and new drug development studies do not involve bone cells such as chondrocytes. Chondrocytes are mesenchymal stem (MSC)-derived cells that produce and maintain the extracellular matrix (ECM) within articular cartilage [48]. Chondrocytes can be cultured by using classical monolayer (2D) techniques; however, it is well documented that chondrocytes under those conditions may lose cortical actin distribution and adopt a fibroblast-like shape instead of classical spheroidal/elliptical shape loss of aggrecan synthesis, and type II collagen, together with an increased type I collagen [49] due to the absence of the specific microenvironment found it in the cartilage [50,51]. Indeed, altered gene expression patterns should be responsible for the changes mentioned above in chondrocytes, which could be absent in the cartilage. There are currently novel strategies for performing chondrocyte culturing on three-dimensional (3D) techniques that support the growth of these cells to a reliable scenario in the cartilage [50,52,53]. As other primary cells, chondrocytes require some growth factors, such as fibroblast growth factor 2 (FGF2), the vascular endothelial growth factor (VEGF), insulin, transferrin, and selenious, that lead to their proliferation and classical expression of markers like type II and IX collagen, aggrecan and the transcription factor Cbfa1 [53,54]. Due to their expensiveness, these specific culturing conditions can limit their use in some institutions.

Despite the challenges related to chondrocyte culturing, some studies have implemented them in several MPS IVA studies [29,36,37,38]. For instance, Sawamoto and Tomatsu, 2019, tested a thermostable keratanase isolated from *Bacillus circulans* KsT202 in 3D-cultured human MPS IVA chondrocytes [29]. Results from those experiments showed the specific KS degradation without affecting other GAGs, such as di-HS-0S, diHS-0S, diHS-NS, or C6S. Likewise, murine MPS IVA chondrocytes were also used for evaluating an AAV-based GT by Alméciga-Díaz et al., 2010. Chondrocytes were transduced with AAV carrying an expression cassette composed of GALNS and/or SUMF1 cDNA under the control of three different promoters: CMV, AAT, and EF1 [38]. SUMF1 encodes for a formylglycine-generating enzyme required for GALNS activation [55]. After four days post-transduction, authors found the highest GALNS expression intra- and extra-cellularly when CMV was placed as a promoter regardless of the presence of SUMF1, providing early evidence of the gene therapy suitability.

Induced pluripotential stem cells (iPSC) have opened new horizons in disease modeling since easy-to-obtain cells, such as the fibroblasts, can be reprogramed as pluripotent stem cells via integrative or non-integrative approaches [56], and then differentiate into any specialized cell, such as chondrocytes. In fact, iPSC-derived MPS IVA chondrocytes were developed from human MPS IVA fibroblasts [37]. Briefly, human MPS IVA fibroblasts isolated from patients with severe and moderate onsets or unaffected were reprogrammed with retrovirus carrying the transcription factors Oct4, Sox2, Klf-4, and c-Myc, which are well-known inductors of pluripotent cells from somatic cells [57]. The resulting iPSCs were used to derivate MSCs and later chondrogenic differentiation. Surprisingly, iPSC-derived chondrocytes establishment was unsuccessful, supporting the need to differentiate chondrocytes from iPSC-derived MSC instead. Under this strategy, authors successfully obtained chondrocytes expressing Col2, aggrecan, and high levels of Col10, suggesting that hypertrophic chondrocytes were achieved. This MPS IVA chondrocyte model was then used for testing a rhGALNS enzyme [37].

Interestingly, after rhGALNS treatment, expression of Col10 was decreased across 25 days of treatment in both severe and moderate MPS IVA chondrocytes, while aggrecan and Col2 surprisingly increased between days 25 and 30 in moderate MPS IVA chondrocytes. Col2 remained unchanged in severe MPS IVA chondrocytes treated with rhGALNS [37]. Likewise, a study conducted by Dvorak-Ewell et al., 2010, tested a rhGALNS enzyme using human MPS IVA chondrocytes isolated from the iliac crest of two MPS IVA patients, whereas unaffected chondrocytes isolated from normal human knee [36]. Even though these models worked well for the in vitro screening of the rhGALNS enzyme, they are inadequate for deep analysis due to their different sources. For instance, chondrocytes’ responses to mechanical forces may vary across anatomical positions [58] and have differential gene expression patterns [59]. Additional MPS IVA fibroblast-, chondrocyte-, and cardiomyocyte-derived iPSC models were successfully developed and well characterized; nevertheless, functional studies are still to be performed in these attractive models [60].

## 3. In Vivo MPS IVA Models

Although in vitro models have helped to understand the molecular and cellular bases of the MPS IVA and screening novel drugs [61,62,63], it is clear that in vivo models provide a more accurate approach [64,65]. Before moving toward clinical trials, animal models are strongly recommended for evaluating the efficacy, pharmacodynamics, pharmacokinetics, and toxicity/safety of new drugs [66]. According to the European Medicines Agency (EMA), the International Society for Stem Cell Research (ISSCR), and the Food and Drug Administration (FDA), the evaluation of advanced therapeutic medicinal products (ATMPs) should be conducted on large animals, such as pigs, sheep, or horses [64], since they provide a most realistic approach. Some findings from small animals, such as rodents, are only partially translated to humans [64,67]. In this section, we describe the current animal models for MPS IVA, their characteristics, and challenges for modeling and drug testing. Table 3 summarizes the significant findings of animal models of MPS IVA.

### 3.1. MPS IVA Mouse Models

The first MPS IVA mouse model was generated in 2003 by Tomatsu et al., who modified embryonic stem cells (ESCs) from mice strain 129SvJ [68]. ESCs were genetically manipulated to induce a cassette’s homologous recombination (HR) using the Cre-LoxP system into C57BL/6 mice. Cre is a site-specific recombinase that catalyzes the HR of known sequences at specific DNA regions (LoxP) [69]. The targeting cassette was designed to disrupt murine *GALNS* by inserting a neomycin resistance (*Neo*^R^) gene under the control of the mouse phosphoglycerate kinase promoter in the intron 1- exon 2 binding regions into *GALNS* gene [68]. The *Neo*^R^ gene was then removed by mating heterozygotes animals with Cre-expressing mice. Consequently, a partial deletion of intron 1 and exon 2 occurred, resulting in a frameshift in homozygotes *GALNS*^-/-^ mice (hereafter called MKC—MPS IVA knock-out—mice). Although this model showed null GALNS activity and classical GAGs accumulation, bone pathology in large bones was unaffected [68]. Such findings were attributed to the absence of KS II in the mouse aggrecan, which supported the lack of any evident skeletal phenotype [68]. We measured mono-sulfated and di-sulfated KS levels in various species. Mono-sulfated KS level is the lowest in wild-type mice (B6C57) among mice, rats, canine, rabbits, cynomolgus monkeys, and humans (lowest to highest in order). Mouse has over 45-fold less mono-sulfated KS level in plasma (20 ng/mL) than humans and 2.5-fold less than rats. In mice, di-sulfated KS is undetectable, while other species have detectable levels. The major findings of MKC mice are displayed in Table 2. An additional MPS IVA mouse model by introducing a missense mutation at cysteine 79 (C79S) using a similar strategy detailed for MKC (hereafter called C2) was also developed [70]. C79 corresponds to the catalytic site in the GALNS enzyme in mice [11]. C2 mice showed similar findings to MKC ones (Table 2). Those models have been used for several assessments, including SDET [29] and classical GT [31,39].

Tomatsu et al. created an MPS IVA mouse model that is tolerant to the human GALNS protein [71]. First, a C79S point mutation was introduced into the mouse *GALNS* gene (mGALNS). Then, a Cre-LoxP-mediated HR was used for introducing into intron 1 an expression cassette containing a human *GALNS* (hGALNS) carrying a missense mutation at the active site (C76S). Blotting experiments from the resulting transgenic *GALNS^tm(hC79S.mC76S)slu^* MPS IVA mice (hereafter called MTOL) revealed the presence of human GALNS transcripts and protein by Northern blot and Western blot, respectively, suggesting that the human cDNA was successfully expressed. As expected, blood and tissue samples from homozygous MTOL mice showed no GALNS activity [71]. Compared to MKC and C2, MTOL showed a marked bone pathology in the growth plate, ligament, and cortical and trabecular bone. Contrastingly, a significant decrease in other sulfatases was noticed in MTOL mice. Since sulfatases require active-site modification by the formylglycine-generating enzyme, encoded by SUMF1 [55], the reduction in the activity of other sulfatases may be related to the hGALNS expression, thereby limiting the availability of SUMF1 for the activation of other sulfatases.

Consequently, the increased GAGs storage in multiple tissues from MTOL mice compared to MKC and C2 may contribute to the observable phenotype of MTOL. Interestingly, the intravenous administration of a hGALNS enzyme in MTOL mice failed to induce any humoral response. In contrast, MKC mice displayed higher levels of anti-GALNS antibodies [71], supporting the success of obtaining a human GALNS-tolerant MPS IVA mouse model. Like MKC and C2 models, MTOL has also been used for testing classical GT [31].

Even though the mouse models mentioned above have been widely used for several studies, mostly related to pre-clinical testing of gene therapies, none developed the classical skeletal display observed in MPS IVA patients. Nevertheless, efforts attempted over the last 20 years with these MPS IVA models led recently to the first clinical trial for evaluating an AAV GT in MPS IVA patients [72]. Hopefully, these pioneer studies will lead to the approval of a new promising alternative to correct bone pathology, which is not currently improved by ERT [21].

### 3.2. MPS IVA Rat Model

Although MPS IVA mouse models have been used for testing several strategies for treating MPS IVA [29,31,39], it is clear that the absence of skeletal phenotype is a significant limitation. Recently, a promising MPS IVA model was developed in rats by introducing a common missense mutation (R388C; equivalent to R386C in humans) in the rat *GALNS* gene using the CRISPR/Cas9 technology [40]. In contrast to mouse models, the rat model showed a reduction of approximately 50% in body weight and 20% in naso-anal length after the first month of life, as well as skeletal alterations, dental malocclusion, fragility, and enamel hypoplasia [40]. MPS IVA rats also showed a significant increase in the KS in the liver, chondrocyte vacuolation, and mitral valve distention. 

This new MPS IVA rat model was used to evaluate an AAV9-based gene therapy by Bertolin et al., 2021 [40]. In those experiments, a codon-optimized rat GALNS cDNA was placed upstream of the ubiquitous CMV early enhancer/chicken β actin (CAG) promoter to reach a widespread expression. Interestingly, the use of 6.67 × 10^13^ vg/kg in four-week-old MPS IVA rats led to the successful AAV9-GALNS transduction in long (femur, fibula, humerus, tibia), flat (scapula), irregular (vertebrae), and sesamoid (kneecap) bones [40]. As expected, a significant GALNS activity was achieved in those bones even six months after treatment with a recovery of the growth plate impairment to wild-type findings. Classical chondrocyte vacuolation in articular cartilage was restored entirely to WT findings in six months-treated rats, suggesting a clarence of accumulated GAGs [40]. Additionally, peripheral tissues such as the liver, heart, and lung were successfully transduced with concomitant trachea and heart pathology normalization [40], supporting the suitability of AAV9 and ubiquitous promoters as promising candidates for the classical GT in MPS IVA. Nonetheless, future assessment in large animals (i.e., non-human primates) must still be addressed to evaluate the long-term safety profile, vector biodistribution, and effectiveness before progressing to clinical trials.

## 4. Future Perspectives

In this review, we have described the current in vitro and in vivo models of MPS IVA, as well as some drawbacks for understanding the pathophysiological mechanisms of the disease and assessment of promising drugs. Major drawbacks include: Skin fibroblasts affected with MPS IVA are commonly used in vitro rather than chondrocytes. However, several canonical states observed on chondrocytes, such as resting, proliferative, and hypertrophic states, and their regulatory pathways [73,74] cannot be assessed in fibroblasts. Likewise, critical chondrocyte functions, such as their involvement in endochondral ossification [75,76,77], cannot be evaluated in skin fibroblasts.Although some studies reported chondrocytes as an MPS IVA model, some consulted works lack critical information regarding cell culturing, including passages and supplements. It is well known that culturing methods greatly influence cell physiology in primary cultures [78].Current mouse models do not display skeletal dysplasia in MPS IVA patients, making it challenging to establish the effectiveness of potential new drugs. Even though Bertolin et al. have shown an improved MPS IVA animal model by using rats able to display some clinical features of the MPS IVA, there is still a high priority for developing large animal models to recapitulate the skeletal dysplasia observed in MPS IVA patients.

In light of these challenges to overcome, we strongly consider that upcoming research should be focused on:A comprehensive characterization of MPS IVA chondrocytes: The full channelome in healthy chondrocytes has been established [79]. Nevertheless, there is no information on this channelome profile in MPS IVA chondrocytes. We also propose conducting functional studies, such as electrophysiological recordings involving passive and active plasma membrane properties and metabolic profile studies, as they can provide new insights beyond the lack of GALNS or GAG accumulation. These premises have been previously addressed in osteoarthritic chondrocytes (OC) and clearly demonstrate differences in OC compared to healthy chondrocytes [79,80].A more realistic microenvironment: Establishing primary chondrocyte culturing by using either iPSC-, MSC-, or surgical specimen-derived sources is strongly recommended. These cultures should be performed in complex culturing systems (i.e., 3D rather than 2D). We also suggest reporting cell culturing conditions, such as passages and supplements, since they can be critical for understanding reported results.Organ-on-a-chip (OoC): Developing novel strategies such as organ-on-a-chip (OoC) is implemented to recapitulate relevant physiological conditions [81,82,83,84], including mechanical stimulation. Surfaceome chondrocyte characterization has demonstrated the expression of several ion channels as a response to biomechanical stress [79,85], and early studies have shown that 2D-cultured chondrocytes affect their response to mechanical stimulation [86,87,88]. Therefore, models based on OoC technologies result in better in vitro MPS IVA models to explore the molecular and cellular consequences of the GAG accumulation in chondrocytes under relevant pathophysiological microenvironments.Large animal models: All current MPS IVA models are derived from the genetic manipulation of rodents (mouse and rat), and there is no evidence of naturally occurring large animal models as described for MPS, such as MPS I, MPS IIIA, MPS IIIB, MPS IIID, MPS VI, and MPS VII [61]. Therefore, the establishment of new large MPS IVA models, such as non-human primates, could be beneficial for the assessment of conventional (i.e., ERT, PC, classical GT) and novel alternatives (i.e., SDET, CRISPR/Cas9-based GT) for treating MPS IVA, since they can provide a larger blood volume, larger tissues, and more human-like anatomy and physiology [89], compared to rodents.

## Figures and Tables

**Figure 1 ijms-24-16148-f001:**
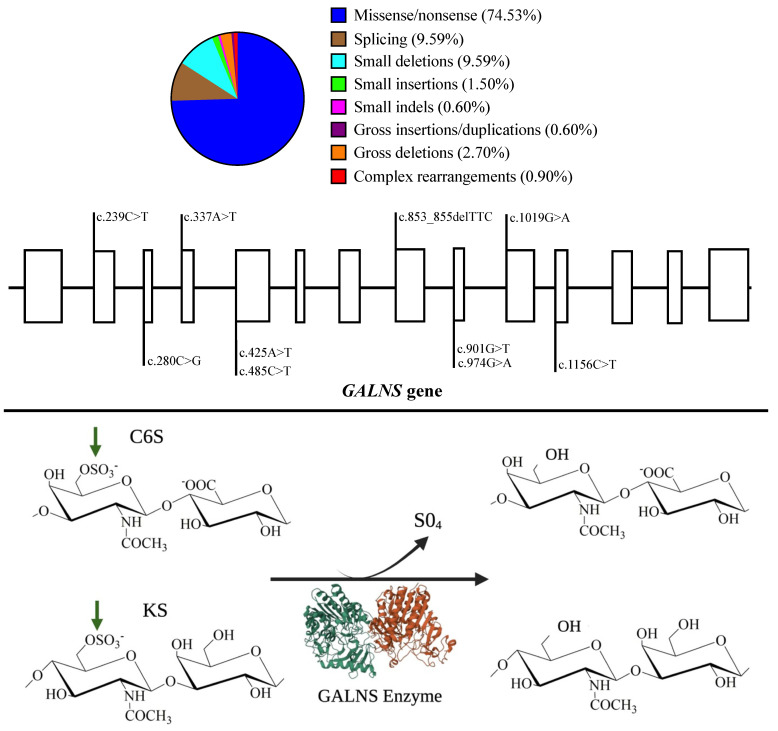
GALNS enzyme activity and the most common mutations on the *GALNS* gene. **Upper.** Global mutation distribution for the *GALNS* gene. Note that missenses are the most common mutations. The most common mutations are displayed along the GALNS gene. Open rectangles represent exons made at scale. Mutations were chosen according to Tomatsu et al., 2005 [15]; Morrone et al., 2014 [13,16]; Cozma et al., 2015 [9]; Tapiero-Rodriguez et al., 2018 [17]; and Pachajoa et al., 2021 [10]. **Bottom.** Schematic representation of the GALNS activity on C6S and KS. Note that the GALNS enzyme (PDB-4FDI) is represented as a homodimer (green and red) that removes sulfate groups (green arrows). Impaired GALNS activity results in the lysosomal accumulation of C6S and KS. Historically, N-N-acetylgalactosamine-6-sulfate sulfatase catalyzes C6S, and galactosamine-6 sulfatase catalyzes KS.

**Figure 2 ijms-24-16148-f002:**
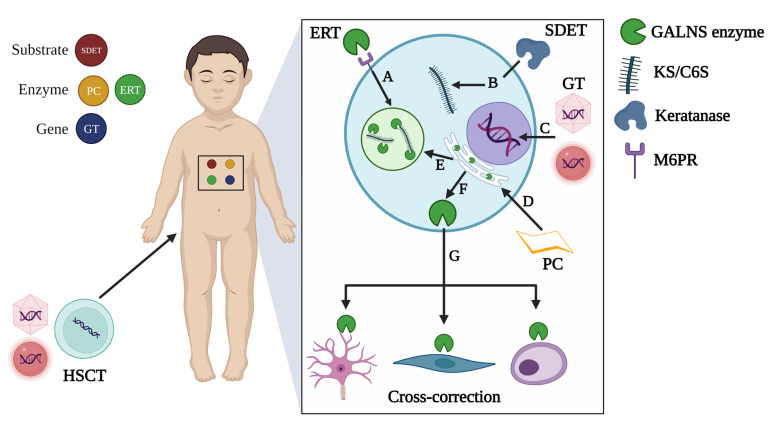
Therapeutical strategies tested in MPS IVA. Strategies can be classified as substrate-, enzyme-, or gene-based approaches. **A.** Enzyme replacement therapy (ERT) leads the uptake of recombinant enzymes via mannose-6 phosphate receptors (M6PR) to be sorted into lysosomes where accumulated substrates will be degraded. **B.** Substrate degradation enzyme therapy (SDET) was evaluated in MPS IVA chondrocytes through incubation with a thermostable keratanase isolated from *Bacillus circulans* KsT202 [29], which specifically degraded KS. **C.** Gene therapy (GT) involves the delivery of therapeutic genes inside the nucleus, which can be attempted by using viral or non-viral vectors as carriers. **D.** Pharmacological chaperones (PCs) contribute to the thermal stability of proteins containing missense mutations that affect their folding. Consequently, stable proteins are sorted to the lysosome (**E**), where they can exert their catalytic activity on accumulated substrates. **F.** Lysosomal enzymes can also be released to the extracellular space (**G**) and taken up by neighboring cells (cross-correction mechanism). This last mechanism is the rationale for HSCT, engineered or not, and GT-based approaches. This figure was created with BioRender.com (accessed on 12 October 2023).

**Table 1 ijms-24-16148-t001:** Overview of current MPS IVA reported models.

Approach	Model	Human	Mouse	Rat
In vitro	Fibroblasts	X	X	
	Chondrocytes	X	X	
	Leukocytes	X		
	iPSC-MSC-derived chondrocytes	X		
In vivo	GALNS knock-out		X	
	GALNS missense mutation		X	X

iPSC. Induced pluripotent stem cells. MSC. Mesenchymal stem cells.

**Table 2 ijms-24-16148-t002:** Cell culture conditions reported for in vitro MPS IVA models.

Purpose	MPS IVA Model	* Plat.	Media	Suppl.	P	Refs.
Proteomics	Human fibroblast	2D	McCoy 5A	10% FBS, 1% P/S	ND	[35]
	Human leukocytes	NA	NA	NA	NA	[34]
ERT	Human chondrocytes	2D	2D: CGM^TM^	3D: Ascorbic acid	ND	[36,41]
		3D	3D: CDM^TM^			
	Human skin fibroblasts	2D	DMEM	15% FBS, 1% P/S	ND	[41,42]
	iPSC-MSC-derived human chondrocytes	3D	MEM	TGFβ3	P1 *	[37]
PC	Human fibroblast	2D	DMEM	15% FBS, 1% P/S	ND	[28]
SDET	Human chondrocytes	3D	CBM™ Basal Medium	NA	ND	[29]
Retrovirus	Fibroblast	2D	DMEM	10% FCS	ND	[43]
	PBLs	2D	IMDM and RPMI	IMDM: 10% FCS, PHA, IL-2RPMI: HS	ND	
	Lymphoblastoid	2D	DMEM	10% FCS	ND	
LV GT	Human fibroblast	2D	DMEM	10% FBS 1% P/S	ND	[33]
AAV GT	Human fibroblasts	2D	NA	NA	ND	[39]
	Murine chondrocytes	3D	NA	NA	ND	[38]
CRISPR/nCas9 GT	Fibroblast	2D	DMEM	15% FBS 1% P/S	P3-P7	[30,32]

* Inferred from the methodology described by authors. AAV GT. Adeno-associated viral gene therapy. CBM^TM^. Chondrocyte Basal Medium (Fisher Scientific). CDM^TM^. Chondrocyte Differentiation Medium (Lonza). CGM^TM^. Chondrocyte Growth Medium (Lonza). DMEM. Dulbecco’s Modified Eagle Medium. ERT. Enzyme replacement therapy. FBS. Fetal Bovine Serum. FCS. Fetal Calf Serum. HS. Human serum. IL-2. Interleukin 2. IMDM. Iscove’s Modified Dulbecco’s Medium. iPSC. Induced pluripotent stem cells. LV GT. Lentiviral gene therapy. MEM. Minimum Essential Medium. MSC. Mesenchymal stem cells. P. Passage. P/S. Penicillin-Streptomycin. PBLs. Peripheral blood lymphocytes. PC. Pharmacological chaperons. PHA. Phytohemagglutinin. Plat. Platform. RPMI. Roswell Park Memorial Institute. SDET. Substrate degradation enzyme therapy. Suppl. Supplements. TGFβ3. Transforming growth factor β3. 2D. Two-dimensional culture/Monolayer. 3D. Three-dimensional culture. NA. Not available. ND. Not disclosed.

**Table 3 ijms-24-16148-t003:** Current in vivo MPS IVA models and their characteristics compared to unaffected animals.

Parameter	Mouse Models			Rat Model
	MKC	C2	MTOL	
*GALNS* gene	* KO	*mGALNS*: C79S	*mGALNS*: C79S *hGALNS*: C76S	*rGALNS*: R388C
GALNS activity	UD	UD	UD	UD
Body weight	UA	UA	UA	50% reduction
Skeletal dysplasia	No	No	No	Yes
Total GAGs	Urine: 6-fold	Urine: UA	Urine: 1.3-fold	NA
	Cornea: 2.3-fold	Cornea: NA	Cornea: NA	
KS	Urine: UD	NA	Urine: NA	Serum: ~3-fold
	Cornea: 1.7-fold		Cornea: UA	Femur: 4-fold
Bone	GP: UA	GP: NA	GP: Irregular structure	GP: Short
	Chon: UA	Chon: Vacuo.	Chon: Vacuo.	Chond: Vacuo.
	Osteob: UA	Osteob: UA	Osteob: Vacuo.	Osteob: NA
	Osteoc: UA	Osteoc: UA	Osteoc: Vacuo.	Osteoc: NA
Kidney EC in glomeruli	Vacuo.	Vacuo.	Vacuo.	NA
Heart Valves	Vacuo.	Vacuo.	Vacuo.	Vacuo.
Liver Kupffer cells	Vacuo.	Vacuo.	Vacuo.	Vacuo.
Other non-GALNS sulfatases	NA	ARSB: Increased	ARSB: Decreased	NA
		IDS: Increased excepting bone	IDS: Decreased	
		Sulfa: Increased	Sulfa: Decreased	

***** KO. GALNS knock-out was achieved by partially deleting intron 1 and exon 2. mGALNS: Mouse *GALNS* gene. hGALNS: Human *GALNS* gene. rGALNS: Rat *GALNS* gene. ARSB. Arylsulfatase B. Chon. Chondrocytes. EC. Epithelial cells. GP. Growth plate. IDS. Iduronate-2-sulfatase. KS. Keratan sulfate. Osteob. Osteoblasts. Osteoc. Osteocytes. Sulfa. Sulfaminidase. Vacuo. Vacuolization. NA. Not available. UA. Unaffected. UD. Undetectable.

## Data Availability

Not applicable.

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
