# Peer review of "Mucopolysaccharidosis IVA: Current Disease Models and Drawbacks"

_ijms, 2023, doi:10.3390/ijms242216148_

Round 1

Reviewer 1 Report

Comments and Suggestions for Authors

Leal and colleagues present reviews of the current in vitro and in vivo models of MPS IVA (mucopolysaccharidosis type IVA) and their limitations. This manuscript holds value for the literature. However, significant editing is needed (see below).

Major Revisions:

1. Add a table categorizing and organizing existing MPS IVA models to aid reader comprehension.

2. In the in vivo models section, consider introducing models by species (mouse, rat, etc.) in separate paragraphs to improve structure.

3. Spell out professional terms (e.g., iPSCs) in full at first use to improve readability.

4. In Figure 1, please provide the percentages for each variant in the pie chart.

Minor Revisions:

1. In line 35, "Even though some mutations have been strongly related to MPS IVA severity, does not exist a clear genotype-phenotype correlation [4, 5]" - Add "It" after "severity" for clarity.

2. In Figure 2, (E), where they can exert their catalytic activity on accumulated substrates. Remove comma after (E).

3. Paragraphs 2.1, 2.2 and 3.1 should not be in italics.

Comments on the Quality of English Language

Moderate editing of English language required

Author Response

Reviewer 1. Leal and colleagues present reviews of the current in vitro and in vivo models of MPS IVA (mucopolysaccharidosis type IVA) and their limitations. This manuscript holds value for the literature. However, significant editing is needed (see below).

Major Revisions:

  1. Add a table categorizing and organizing existing MPS IVA models to aid reader comprehension.

Response. We appreciate this suggestion and have included a new table showing an overview of the current MPS IVA-reported models. Please see lines 75-78.

  1. In the in vivo models section, consider introducing models by species (mouse, rat, etc.) in separate paragraphs to improve structure.

Response. We acknowledge this suggestion; nevertheless, our current structure includes separate sections for mice (section 3.1) and rats (section 3.2), which we consider accurate.

  1. Spell out professional terms (e.g., iPSCs) in full at first use to improve readability.

Response. We have proofread the paper and spelled out any missing terms. The reviewer will clearly identify as red font words (i.e.: Please see lines 28 and 35…)

  1. In Figure 1, please provide the percentages for each variant in the pie chart.

Response. We appreciate this comment. After considering adding full percentages to the pie chart, we noticed that it would be a hard-reading pie. Nevertheless, we have included them in front of each identification color legend. Please see Fig. 1.

Minor Revisions:

  1. In line 35, "Even though some mutations have been strongly related to MPS IVA severity, does not exist a clear genotype-phenotype correlation [4, 5]" - Add "It" after "severity" for clarity.

Response. We have fixed it. Please see line 37.

  1. In Figure 2, (E), where they can exert their catalytic activity on accumulated substrates. Remove comma after (E).

Response. We have fixed it.

  1. Paragraphs 2.1, 2.2 and 3.1 should not be in italics.

Response. We have removed italics from subheading sections.

Reviewer 2 Report

Comments and Suggestions for Authors

This review discusses the disease models for Mucopolysaccharidosis IVA (MPS IVA), a rare genetic disorder caused by mutations in the GALNS gene. This disorder leads to the impaired degradation of C6S and KS in the body, causing skeletal abnormalities and non-skeletal manifestations in patients. Both in vitro (fibroblasts, chondrocytes, and lymphoblast) and in vivo (mouse and rat) models have been developed to understand MPS IVA and evaluate potential treatments. However, these models have limitations, such as not fully representing the bone-related issues observed in MPS IVA patients. Specifically, chondrocytes, the primary cells affected in MPS IVA, have not been extensively studied, and existing animal models do not perfectly mimic the human skeletal dysplasia. The manuscript reviewed current MPS IVA models and evaluated the limitations, highlighting the need for improvement and potential new research approaches. The review is well composed for assessing various models for MPS IVA. The references are up-to-date and relevant.  The authors could consider including 1) Comparing proteomic results from fibroblasts and leukocytes,  and their value to the understanding the disease in MPS IVA patients. 2) Discuss potential differential catalytic properties between human and mouse GALNS. The findings from mouse models suggest that there might be a critical level (or threshold) of KS and C6S, below which the enzymatic activity of GALNS is sufficient to manage the degradation effectively, for human and mouse GALNS enzymes.

Author Response

Reviewer 2. This review discusses the disease models for Mucopolysaccharidosis IVA (MPS IVA), a rare genetic disorder caused by mutations in the GALNS gene. This disorder leads to the impaired degradation of C6S and KS in the body, causing skeletal abnormalities and non-skeletal manifestations in patients. Both in vitro (fibroblasts, chondrocytes, and lymphoblast) and in vivo (mouse and rat) models have been developed to understand MPS IVA and evaluate potential treatments. However, these models have limitations, such as not fully representing the bone-related issues observed in MPS IVA patients. Specifically, chondrocytes, the primary cells affected in MPS IVA, have not been extensively studied, and existing animal models do not perfectly mimic the human skeletal dysplasia. The manuscript reviewed current MPS IVA models and evaluated the limitations, highlighting the need for improvement and potential new research approaches. The review is well composed for assessing various models for MPS IVA. The references are up-to-date and relevant.

The authors could consider including:

1) Comparing proteomic results from fibroblasts and leukocytes and their value to the understanding of the disease in MPS IVA patients.

Response. In section 2.1, we have linked the relationship between proteomic findings in fibroblasts and their correlation with the pathophysiological events of the MPS IVA. For instance, it is clear that impaired lysosomal function affects mitochondrial homeostasis, explaining the pro-oxidant profile noticed in patients (See lines 114-120). Regarding leukocytes, we have included the following: In fact, authors also found downregulated several enzymes linked to Kreb’s cycle, providing further evidence of the mitochondria-lysosome pathway disturbance in MPS IVA leukocytes as described for MPS IVA fibroblasts [31]. These findings should drive upcoming research to understand MPS IVA not only as a lysosomal-affecting disease but as a cell-homeostasis-affecting pathology (See lines 148-153). We really hope this review will encourage basic researchers to explore these interesting findings to have a full picture of the MPS IVA.

2) Discuss potential differential catalytic properties between human and mouse GALNS. The findings from mouse models suggest that there might be a critical level (or threshold) of KS and C6S, below which the enzymatic activity of GALNS is sufficient to manage the degradation effectively for human and mouse GALNS enzymes.

Response. Even though we agree with the reviewer regarding potential catalytic differences between human and mouse GALNS, the absence of KS II in aggrecan molecules in mice is mainly responsible for the lack of an evident skeletal dysplasia, as we discuss in section 3.1 (Please see lines 247-250). Besides, we have described the following: We measured mono-sulfated and di-sulfated KS levels in various species. Mono-sulfated KS level is the lowest in wild-type mice (B6C57) among mice, rats, canine, rabbits, Cynomolgus monkeys, and humans (lowest to highest in order). Mouse has over 45-fold less mono-sulfated KS level in plasma (20 ng/ml) than humans and 2.5-fold less than rats. In mice, di-sulfated KS is undetectable, while other species have detectable levels (See lines 250-255).

Round 2

Reviewer 1 Report

Comments and Suggestions for Authors

Leal and colleagues present reviews of the current in vitro and in vivo models of MPS IVA (mucopolysaccharidosis type IVA) and their limitations. This manuscript holds value for the literature. However, a little editing is needed (see below).

Discussion

1.      In the conclusion, the authors should expand on the major limitations of the current MPS IVA disease models, both in vitro and in vivo, to clearly highlight the gaps in modeling this disease.

2.      Following the limitations, the authors should provide specific recommendations on which kinds of models or techniques should be the priority for future development and validation to address these limitations. Listing 2-3 key next steps would strengthen the conclusion.

Overall, this is a well-written review providing helpful analysis of the MPS IVA modeling landscape. Addressing the limitations and future directions in the conclusion will improve it further. I recommend this paper for publication pending minor revisions. Please feel free to contact me if you would like me to clarify or expand on any of my comments.

Comments on the Quality of English Language

Minor editing of English language required

Author Response

Reviewer 1. Comments and Suggestions for Authors

Leal and colleagues present reviews of the current in vitro and in vivo models of MPS IVA (mucopolysaccharidosis type IVA) and their limitations. This manuscript holds value for the literature. However, a little editing is needed (see below).

Thank you very much for your kind comments.

Discussion

  1. In the conclusion, the authors should expand on the major limitations of the current MPS IVA disease models, both in vitro and in vivo, to clearly highlight the gaps in modeling this disease.

Response. We appreciate this suggestion and have re-written down the future perspectives to make more clear the current drawbacks and upcoming research for overcoming them. Please see lines 321-374.

  1. Following the limitations, the authors should provide specific recommendations on which kinds of models or techniques should be the priority for future development and validation to address these limitations. Listing 2-3 key next steps would strengthen the conclusion.

Response. We appreciate this suggestion and have re-written down the future perspectives to make more clear the current drawbacks and upcoming research for overcoming them. Please see lines 321-374.

Overall, this is a well-written review providing helpful analysis of the MPS IVA modeling landscape. Addressing the limitations and future directions in the conclusion will improve it further. I recommend this paper for publication pending minor revisions. Please feel free to contact me if you would like me to clarify or expand on any of my comments.

Thanks a lot for your positive comments.